# ARE MACHINES AUTOMATING MORALITY?

## ABSTRACT

The advent of artificial intelligence (AI) and machine learning has ignited a profound inquiry into the morality of machines. In a quest for efficiency, pleasure, comfort, we delegate and automate more and more decisions and actions to AI-based systems. In this paper, we delve into the complex interplay between artificial intelligence and morality. We thus address the fundamental question of whether machines possess morals and if machine learning systems can learn about moral values. As automated systems increasingly take on decision-making roles in our lives, ethical concerns are growing among researchers and philosophers. Making an ethical decision has always been connected to human agency. We try to highlight the prevailing utilitarian ethics found in the tech-centric Silicon Valley culture and its influence on the development of such automation. As machines make more and more decisions, they consequently express a certain morality. In this paper we highlight the emergence of the idea of "moral machines" to describe machine learning systems, for instance in the context of autonomous vehicles, where AI-based systems must take ethically challenging decisions - we thus discuss the pertinence of the well-known "trolley problem" as an illustrative example to explore the utilitarian aspect of these ethical dilemmas, it applies to any domain where machines make moral choices based on patterns and data. Calling those machines "moral" underline the fact that AI systems make moral choices without any human intervention. However this term is not confined to autonomous vehicles. This paper examines the implications of this automated morality and how it can affect individuals' sense of responsibility raising the questions about the future of morality. Automated values challenge the idea of responsibility and moral agency. We then call for a thoughtful and critical examination of ethical and political implications of automated systems shaping our moral background. In the age of technological disruption, ethical questions surrounding automated morality must be addressed to safeguard our ethical compass.

## 1 INTRODUCTION

Artificial intelligence (AI) and machine learning are reshaping the landscape of our daily lives. As they do so, questions emerge: Do machines have morals? Can they learn about morality? And what would then be the consequences of this automatically-produced morality? As we increasingly delegate decision-making tasks to AI-based systems, concerns about the ethical implications of these systems have gained importance among researchers and philosophers. Some have come to the conclusion that it may be impossible to achieve genuine morality in a machine (Coeckelbergh, 2010).

We start by introducing the key difference between "morality" and "ethics". Derived from the Greek word "ethos" meaning "way of life; habit; manners; state of mind", ethics is a branch of philosophy concerned with human behavior and, more specifically, the conduct of individuals in society. On the other hand, "morality" has a roman etymology of the same word (from the latin "moralis" meaning "relating to manners"). In everyday life, the two words are synonymous. However, ethics, unlike morality, are situated in the conduct of action and not in the action itself. Actions that can be judged as morally wrong can be carried out for ethical reasons (Garcia, 2016). We could say that morality is a set of obligations to be respected if one does not want to get into trouble with the law, colleagues or relatives.

Morality is by essence an universal judgment. Ethics are more personal, and have to do with one's behavior. In addition, ethics can be specific to an industry and depend on the goal one is pursu-

ing. We will use both terms in this paper. We consider here that the ethics of automated systems represents the way to achieve a behavior considered as "moral" with regard to the considerations of various human societies. Historically, "moral universalism" is opposed to "moral relativism", according to which it is important to be able to consider that morality is also relative to a society, a region of the world, a religion... We will not enter into this debate here. However it seems important to mention that the notion of universalism is at the core of many questions in moral philosophy. Historically, the French Revolution's *Declaration of the Rights of Man and of the Citizen* was universal in scope (Sepinwall, 2005). The French signatories of the time considered that the moral values defended by this Declaration were not intended to be called into question or considered relative to France at the end of the 18th century.

Another distinction concerns the "absolute" or "non-absolute" character of a moral law. It is possible to consider that the purpose of the moral law is to be able to be applied for all and at all times, whatever the consequences (Kant & Korsgaard, 1998). Some moral traditions, such as utilitarianism are universalists as they tend to be applied universally but are non-absolutist since they are concerned with the context and the consequences of an action (Salvat, 2020).

In this paper, we focus on the moral values expressed - or not - in the automatic decisions made by a model – whether we consider "algorithms", "AI-based systems", "machine learning systems". These three terms cover different meanings, but share the reality of a more or less autonomous automation of various human processes. While the principle of algorithms is an established feature, the particularity of machine learning is that a model now has the ability to automatically learn patterns not defined manually, and on such a scale that it becomes uninterpretable by its designers (Campolo & Crawford, 2020). The question of model interpretability has become a pressing issue as learning methods have become more complex, to the point of creating what are known as "black boxes" , i.e. models whose inner workings are obscure and opaque. We are aware that not all existing models are affected by this phenomenon, although many of today's most popular models are.

Our starting point is an observation : at every moment, at every decision, we, as moral agents, express our moral principles. If humans and the world were as they should be, there would be no need for morality. It is then based on the discrepancy between how things are and how they should be. Morality is therefore based on the assessment of this disparity, which is only possible within this gap between reality and what an individual can wish for. Questioning morality consists in evaluating whether the individual thinks that its action, judgment, can be made universal (Kant & Korsgaard, 1998). Hence, questioning whether our actions are morally good, refers to the idea of universalisation and of the "absolute" quality of our personal ethical preferences. Ethics theories were developed at times when philosophers assumed that individuals - as a group or a sole person - were the main - if not the only - moral agent (Gordon & Nyholm, 2022). However, nowadays, we rely on machines, and we can expect to depend even more on machine learning intelligence in the future. Therefore, we, as individuals delegating part of our decision-making process, need to be confident that the decisions that are supposedly being made on our behalf are actually beneficial to our well-being.

To question the "morality of machines" is to ask whether the decisions "made" by them - we can consider any decision resulting from an automated process - reflect moral values, and whether "learning machines" can be considered as moral agents defending particular moral values. The goal of this paper is then to study how machines learn about morality and thus to exhibit the current and future automation of our moral standards. Being more and more autonomous, machines will require some kind of a "moral framework to guide their action" (Rubin, 2011, pg. 51). The moral question here is therefore closely related to interpretation, and the possibility of understanding how a model can produce a certain moral result.

The current issue arising from machine learning is whether these "machines" could autonomously develop their own ethical values. We would like to point out the limits of this argument: as a matter of fact those machines automate morality but some researchers argue that they cannot take any ethical decision as they ignore what an ethical debate and an ethical decision are (McDermott, 2020). They are not consciously making a decision. Those machines could be compared to an individual ignoring the causes that push her or him to act in a certain way and therefore expressing some moral preferences. They are rather the representations, the traces, of their origin and of the goals pursued by their creators and the organizations that use them.

We proceed as follows. First, we wish to take a step back on moral dilemmas in order to look at what philosophy says about ethics and morality, in particular in our modern era. We then highlight the rise of utilitarian ethics within the technological landscape and the ethical concerns this raises. Introducing the notion of "moral machines", we examine how AI-based systems appear to de facto automate moral choices, particularly in the context of autonomous driving scenarios. Our discussion then extends to the challenges and disruptions this automation causes. We seek to highlight the ethical implications of decisions made by those systems, and conclude by addressing the political and democratic dimensions inherent in the automation of morality. Our aim is to encourage in-depth reflection on the societal implications of technological innovation, while seeking to show the role of ideological, economic and political motivations behind these processes.

## 2   A BRIEF HISTORY OF MORALITY

The history of morality is deeply linked with the history of philosophy. Morality has been a food for reflection and debate from the Greek philosophers 2,500 years ago all the way to Enlightenment thinkers and our modern times. For many philosophers, morality was the goal of the human being. Despite a diverse array of theories and approaches, philosophers have all tried to understand and display the foundations of morality, the criteria for ethical behavior, and the implications of moral principles for human conduct and society. According to the Kantian categorical imperative first developed by Kant in *Groundwork for the Metaphysics of Morals*, "act only in accordance with that maxim through which you can at the same time will that it become a universal law" (Kant & Korsgaard, 1998, pg. 31), an action would be morally good if, and only if, it could be universal and absolute. Kantian ethics are often a very useful tool to solve moral dilemmas as their goal is to find a supreme principle of morality. Kant's idea of universalism and absolutism of moral action, of the good deed, foreshadows the automatic conception of morality developed by automated systems. By definition, morality is meant to be universal and absolute and therefore automatable. Following Kant's maxim, we could argue that this is the only way to guarantee its value. An action is moral if it is automatable. As a result, designing learning machines partly comes down to wondering whether the maxim of the machine's action can be universalized to determine its moral value. This moral issue is comparable to the broader quest of machine learning research for a generalizable model.

This is morality in the strict sense of the term. However, philosopher David Hume explains that the approval of acts that conform to the rule induces a second obligation, which he calls moral obligation, with its own additional force (Saltel, 2019). Hume's originality lies in linking all systems of social and political obligation to the goal of the proper execution of justice. This is true of the most important of them, allegiance to government, but it is also true of other systems of artificial morality, established for the convenience of social life, thereby consolidating property relations through the manifestation of good intent: constancy in friendship, "good manners", but also the particular rules of such and such community or corporation.

For most part of the history of moral philosophy, human beings were considered the only moral agents, thus the only ones responsible for their actions and decisions. However, the rapid deployment of automatic decision-making solutions based on machine learning is transforming this historical consideration. An individual's decision is the mirror image of his or her moral convictions. As a result, the more so-called artificial intelligence solutions are used in place of individuals, the more it will be possible to consider these systems as moral agents.

**An utilitarian AI**

French philosopher and theologian Jacques Ellul showed that modern technology is characterized by certain properties. Firstly, it tends to become independent with regard to all norms, like traditional morality, to hinder its mathematical march towards its result. Technology, in a broader sense, loves nothing nor respects anything, but it has only one role which is to strip away and to clarify. The aim of technology is to rationalize and to transform everything into a means (Ellul, 1988). Sacralized by human beings, technology has ceased to be a means to an end and has become an end in itself (Ellul, 1980).

Technology is often described as neutral. However, while it's mistaken to regard technology as such (Stinson, 2022) - every means implicitly serves an end and carries a certain vision of the world and how it should be, "the essence of technology is by no means anything technological" (Heidegger,

1977, pg. 4) - it is interesting to note that it is closely linked to the morality of modernity. Technical progress has been part of the history of modernity since the scientific and philosophical revolution that began in the 16th century. From then on, the simple fact of developing ever more sophisticated technical tools, with the proclaimed aim of eliminating suffering, improving human life and even augmenting the actual human being, is part and parcel of a certain form of morality. Modernity is perpetual change and growth. It is the "tradition of the new" (Rosenberg, 1994).

Strongly established in the Anglo-Saxon tradition since Jeremy Bentham in the 18th century, consequentialist morality, and more specifically utilitarian ethics, has dominated debates and infused current technological developments (Salvat, 2020). As a result, utilitarianism has become the dominant morality in Silicon Valley and, by extension, in the innovations that are emerging there. From this point of view, the aim of technology would therefore be to eliminate all suffering and produce the most rational and objective results possible. The moral value of an action would then depend exclusively on the result of the action. From this point of view, utilitarianism is anti-Kantian. Consequentialist morality objects to Kant that there is no such thing as the good in itself. Everything depends on the usefulness of a particular action in a particular context.

The increasingly widespread delegation of our decision-making process to machine learning solutions has real-world ramifications in various industries. The ubiquity of artificial intelligence in our lives is undeniable, affecting sectors such as law, healthcare and finance, while influencing our personal and professional choices (Schrage, 2020). However, this expansion has concentrated power in the hands of a few digital giants (Campolo & Crawford, 2020). Yet the moral consequences of decisions - partly or fully - delegated to AI systems is a pressing issue, as there is concern that they may disproportionately benefit or harm certain social groups (Barocas & Selbst, 2016). For example, credit applications are often biased, favoring certain demographic groups and disadvantaging others (O'neil, 2017). Even worse, an algorithm used in court to "predict" recidivism turned out to be racist (Angwin et al., 2022). The utilitarian perspective is no stranger to these excesses. By design, a model is utilitarian in the sense that it is built with an objective in sight. Consequently, it adapts its functioning to this objective. Discriminatory consequences stem from the failure to define such an objective. If we take the example of the algorithm used to predict the risk of recidivism, its objective, "by design", was not to be non-racist, but to provide the most accurate estimates. Racist bias is an unintended but real consequence.

## 3  MORAL MACHINES

The culmination of this utilitarian morality is materialized in the evolution of how a decision is made. Machine learning models are supposed to support human management processes and in many cases replace human decision-making. When indicators and other numerical data are no longer used to advise decisions, but to make those decisions, the concept of decision-making changes. The idea is not to entrust everything to human intuition, whose flaws are well documented (Fry, 2018), but to avoid falling into a "governance by numbers" (Supiot, 2015) where the use of numerical data considered impartial by nature - despite the numerous cases of discriminatory bias that constantly remind us of the fragility of such systems (Brous & Janssen, 2020) - becomes the norm. Nevertheless, such indicators are increasingly used today, including to modify and influence the behavior of the agents they are supposed to describe. This pushes societies towards an "algorithmic governmentality" (Rouvroy & Berns, 2013). Applied to the automation of moral systems, we can say that we are facing an automated algorithmic morality.

In particular, this is what happens in autonomous driving. Since we may no longer be responsible for the actions of autonomous vehicles, for example, machine learning systems would make these moral choices for us using an utilitarian framework. The aim of any autonomous vehicle is to replace the human act of driving. Thus, human decision-making is being replaced by machine learning systems. A common way to explain and present the ethical dilemmas that autonomous vehicles are facing according to many researchers and philosophers is the well-known "trolley problem". Formulated before AI, trolley problems in AI come in all shapes and sizes, and while decisions don't necessarily have such fatal consequences, the actions AI makes can cause problems for individuals, and society as a whole (McKendrick & Thurai, 2022). Comparing ethical dilemmas taking place for real with the philosophical and psychological example of the Trolley problem is quite controversial among philosophers who have written about this (Nyholm & Smids, 2016). However it does offer a pretty

good general illustration of what is an ethical dilemma and what is an ethical decision despite its weakness, for instance because we tend to compare how a human would react in front of a panicking situation and an autonomous vehicle, whereas we can consider that it will not react the way a human being would react in such scenario.

This "problem" tends to test the utilitarian side of the surveyed individuals. In the case of autonomous cars, it is necessary to develop an artificial driving system, that is to say a software that is used to make driving decisions for the car while the driver is doing something else - at least not driving. Should an autonomous car give priority to humans over pets, to passengers over pedestrians, to more lives over fewer lives, to women over men, to the young over the old, to the healthy over the sick, to higher social status over lower social status, to honest citizens over lawbreakers? Finally, should the car swerve (action) or stay on course (inaction)? All of those theoretical dilemmas tend to test our sense of morality.

Autonomous vehicles are frequently referred to as "moral machines", a term used in a study conducted by researchers at the MIT Media Lab (Awad et al., 2018). The website created for such research in 2014 - moralmachine.net - lists all cases of dilemmas that an autonomous car would face in case of a malfunction of the driving system enforcing it to "make" a choice. As those AI-based systems act, they play out a certain morality, thus automating moral choices. Therefore, we could consider that they have morals. We will show later on that this claim may be an overstatement.

## 4   THE INDIVIDUALS ARE UNCONSCIOUS OF THE PHENOMENON

We argue here that individuals struggle to really consider what it means to automate morality when it comes to automated systems. Is it the end of human morality ? According to Jurgen Habermas (2014, pg 56), "morality will ensure the freedom of the individual to lead his own life only if the application of generalized norms does not unreasonably lace in the scope for choosing and developing one's life-project". Automating morality therefore has an impact on people's perception of freedom, but also on their perception of responsibility. "Algorithms provide a kind of convenient source of authority. an easy way to delegate responsibility; a shortcut that we take without thinking" (Fry, 2018, pg. 18). However, the ability to hold each other accountable for our moral choices is a crucial consideration in the functioning of society. Habermas (2014) argues that individuals who engage in moral judgment and action acknowledge complete accountability toward one another, ascribe the ability for self-governance to both themselves and others, and anticipate mutual solidarity and equal respect. The individual must be morally responsible for his or her life choices (Cohen, 2017).

Technical developments and the increasingly widespread use of probabilities in organizations - which is what machine learning systems are based on today - initially lead modern times to a form of "reflexive madness" (Sloterdijk, 2006), permanently short-circuiting the psyche. Now, faced with digital technical innovations, this reflexive madness is becoming a "new ordinary madness" (Stiegler, 2016). French philosopher Bernard Stiegler argues that our modern society is a society of disruption, a term he defines as what moves faster than any will, whether individual or collective. The aim is to disrupt everything, to go faster and faster. The result is a never-ending chase between innovation, which constantly creates legal loopholes, and the law. This cult of radical, disruptive innovation results in a kind of dissolution of the State, which must be constantly "reformed". This loss of reason is precisely the strongest attack on the principles of modernity and independence of thought. Alienated by technology, individuals sink into madness. It is as if we were delegating our moral responsibility for the simple reason that we may no longer be fully capable of exercising our own moral sense. This loss of morality as a guide to our actions, replaced by efficiency, novelty and profit, is a loss of our human reason (Stiegler, 2016). Human morality ends up copying the moral values represented by machine learning-based systems. Theodor Adorno and Max Horkheimer already announced in 1944 that the mathematical process has been transformed, so to speak, into a ritual of thought (see (Horkheimer & Adorno, 2002) for more information).

## 5   DISCUSSION

In this paper, we have attempted to provide an overview of the moral and ethical issues raised by the increasing presence of automated systems in various decision-making processes. In what follows,

we will focus on the changes this brings about, as well as on what we consider to be real limits to the consideration of an "automatic morality" specific to machines.

## 5.1 NEW MORAL AGENTS

For most part of the history of moral philosophy, human beings were considered the only moral agents, thus the only ones responsible for their actions and decisions. However, the rapid deployment of automatic decision-making solutions is transforming this historical consideration. An individual's decision is the mirror image of his or her moral convictions. As a result, the more so-called artificial intelligence solutions are used in place of individuals, the more it will be possible to consider these systems as moral agents.

In machine learning, models learn automatically from the discovery of patterns, recurring motifs in the data sets they process. As such models are used to support and guide human decisions, they learn from the patterns they notice. Automatic morality is not just about autonomous vehicles. In all areas where we wish to delegate decisions to automatic systems, we will end up with identical moral consequences. Which resume to choose between two candidates? To whom should we give priority for a bank loan? Who should we vaccinate first? Who should receive state aid? Who should die in a car accident? With each moral question addressed by automated systems, we delegate part of our moral authority, especially when it becomes difficult to interpret the outcome. In the case of black boxes, the potential uninterpretability of the decision (Szegedy et al., 2014), and therefore the loss of understanding, leads to a sort of de facto moral autonomization of the machine. Consequently, the more opaque the functions of an automated system, the less explicable its results.Nowadays, neural networks with millions of parameters are very difficult to interpret. Instead, they can be justified a posteriori.

We would like to draw attention to one particular risk of this automation. Indeed, delegating moral values means no longer wishing to change a society's practices; it means considering that from now on, these choices are no longer choices at all, and become a metric of the technical and technocratic environment in which individuals live. Automated systems extend bureaucracy.

We have tried so far to display that moral values are becoming automated in certain specific sectors. Does this open the possibility of a general automation of morality? Or even to the extension of algorithmic morality to human morality ? In the future, individuals could become more and more dependent on automated systems making moral decisions for them. Therefore, this automated morality could become the only moral values represented within society. Individuals may no longer have a choice of values, since this automated morality would have become sufficiently widespread for individuals to have little choice but to model themselves on it. Sometimes, a technological tool can even end up encouraging behavior that runs counter to the values and norms it was designed to promote. For example, the answering machine was originally designed to make the individual more available, but in practice it is used to make oneself unavailable when desired (De Mul, 2009).

## 5.2 THE ISSUE OF FREEWILL

Transferring some of our decision-making processes to automated systems means assuming that the only goal of all decisions is to apply strict so-called rationality, more precisely algorithmic rationality. Since the scientific and philosophical revolution of the 17th century and philosophers of the Enlightenment, we have built our political systems on the basis of freedom, progress, justice and rationality (Strauss, 1975). We could then believe that AI-based systems could achieve this goal.

However, even if the question of free will is not entirely our subject here, we feel it is necessary to point out that the adoption of moral actions requires free will and independence of thought - something a machine obviously lacks. "For a machine to know that a situation requires an ethical decision, it must know what an ethical conflict is" (McDermott, 2020, pg. 7). Agents (individuals or machines) must have free will precisely because they must make decisions in order to be ethical. It is also a question of responsibility. Assuming free will means that an agent is responsible for a decision or an action. Therefore, even if machine learning systems outperform humans in pursuing maximal utilitarian optimization, it may be impossible to build truly moral agents without consciousness (Anderson & Anderson, 2007).

## 5.3 ARE MACHINES REALLY CAPABLE OF MORAL CHOICES?

Automating morality means that potentially the whole world would share the same morals, or at least the same moral machines. This raises the question of the generalizability of machine learning moral models. Blaise Pascal (2007, 294) famously said "A strange justice that is bounded by a river! Truth on this side of the Pyrenees, error on the other side." in order to highlight the cultural differences regarding truth and the notion of justice. Pascal had taken his inspiration directly from Montaigne's *Essays*, who was, in his eyes, the most coherent radical skeptic philosopher. Indeed, Montaigne (1993, pg. 653) had already formalized something similar: "What kind of truth can be limited by a range of mountains, becoming a lie for the world on the other side!". Again with Montaigne (1993, pg. 231), we can even broaden the subject of cultural confrontation: "every man calls barbarous anything he is not accustomed to". Morality appeals to the deep soul of peoples, places and countries and seems very contextual. What do machines consider barbaric? The composition of the database has a major influence on the results of machine learning systems. How will a machine behave when confronted with out of distribution "moral data" - usually from countries or populations underrepresented in datasets? Will it force the application of Western morality in a foreign country? Or will it lose all sense of morality?

The problem is therefore quite acute today. The "moral machines" study conducted by MIT reveals something fundamental about morality around the world, and how difficult it could prove to propose automatic morality on a global scale (Awad et al., 2018). To the question, "Should an autonomous car spare a baby or an elderly person?", East Asian countries such as Japan, China and South Korea clearly answer that the elderly person should be saved. Here we can see the Confucian influence and the respect due to elders, which is very strong in these cultures. In contrast, Western countries answer in the average range, or clearly in favor of sparing the baby. These individualistic cultures, which accord equal dignity to all individuals, favor babies who have their lives to live. These results show that despite globalization, despite worldwide interconnection, ethical and moral preferences differ greatly between civilizations, and remain rooted in their cultural matrices (Confucianism, Christianity, etc.). These findings also have very concrete ethical consequences: Should Volkswagen or Tesla program their autonomous cars differently depending on the country, so as to spare the young in France and the elderly in China?

The large-scale development of autonomous vehicles in fact faces the problem of the complexity of decision-making, due in part to the persistence of human actions that are statistically unavoidable yet difficult to predict. We need to distinguish between an autonomous vehicle operating in an open environment and the automatic metro lines that have been operating for several years in closed environments, most often in underground tunnels. In reality, in all sectors where automatic systems are used and exert a greater or lesser influence on the decision-making process, they always do so in accordance with - more or less exhaustive - predefined rules. It is therefore essential to carefully examine those environments in which sufficient control has been established to authorize the integration of automatic systems. Forecasting all scenarios is crucial due to the need to manage the potentially irrational aspect of human behavior (Hu et al., 2019) as well as challenging weather situations that can affect sensors (Zang et al., 2019) or which, in the case of a human driver, would involve a violation of the rules laid down for ordinary situations.

Even if we imagine a well-disciplined population respecting all the rules, it's impossible to eliminate total unpredictability. Machines have great difficulty in making instantaneous and improvised decisions, just as individuals can react in a matter of seconds in critical situations. This introduces an element of uncertainty that machines cannot handle the same way (Nyholm & Smids, 2016). The freedom of decision available to individuals is a dimension that is absent in the field of automated systems. In the case of indecision, which can be a decision in itself, machines do not possess the ability to recognize this option, or at least they are not given the opportunity to possess it.

This area of uncertainty, which is difficult to grasp, raises questions about the ability of AI to make autonomous, moral decisions outside a fully controlled environment in which moral issues do not arise. It suggests that the only way for an automated system to make genuine moral decisions would be if it could make them in an open, uncontrolled, environment. Arguing that machines automate morality means forgetting the human will - ideological, political, economic - at work in their design and use.

This moral automation is thus the work of individuals who design environments which encourage control. This control can be simply a physical one, as in an underground subway system, but it can also be political, on the scale of an entire society. We have argued that a controlled environment was necessary to implement an automated system. In the opposite direction, the establishment of an automated system can help with controlling an environment. On this matter, the Chinese example of the social credit system demonstrates the totalitarian risk present in this technical control (Creemers, 2018).

## 5.4 MORALS ARE POLITICAL: WHO DEFINES MORALITY?

We must never forget that all these models need data to train and operate. A recent study by six researchers from different universities suggests that training data may eventually be missing (Villalobos et al., 2022). This would imply a limitation in the learning of moral values. This type of research reveals the fragile nature of automation, regardless of its nature. Automated systems depend on the ability of models to process large quantities of data, and therefore on the capacity to harvest them. The question then arises of the limits that governments, international institutions and parliaments may or may not impose.

Therefore, this debate - and, more generally, all questions relating to technological innovations - can be summed up in Shoshana Zuboff (2019)'s three questions: What to decide? Who decides? Who decides who decides? These are eminently political and moral questions. The practical problems arising from the automation of morality are therefore related to political and democratic issues.

The utilitarian injunction to consider only the result of an action encourages us to focus solely on the superficial, directly visible results of technological advances, without seeking to study them in detail. Silicon Valley's new mantra is WYSIWYG: "what you see is what you get" (Alloa & Soufron, 2019). This suggests a preference for immediate, visible results over a comprehensive understanding of the wider implications, ethical considerations and potential consequences associated with these innovations. By brandishing WYSIWYG as its sole political horizon, Silicon Valley delegitimizes in advance any critical reflection on technological innovations and the meaning attached to them. In essence, Silicon Valley's utilitarian vision raises concerns about the potential consequences of a technological culture that privileges immediate, observable results to the detriment of in-depth reflection on the ethical, social and cultural dimensions of innovation. Except that today, a large proportion of technological innovations and automated systems emerge from this technological and economic environment. The utilitarian principle should therefore not be the only guide, insofar as it prevents deeper reflection on the functioning and the future of technology. Therefore, it prevents asking political questions.

However, the political question has always been present in the technological environment, and in Silicon Valley in particular. If it is a geographical location, it is also a place of political and ideological influence and power. In 1996, British sociologists Andy Cameron and Richard Barbrook spoke of a "Californian ideology" in which left-wing ideas and economic liberalism converge Barbrook & Cameron (1996). The Silicon Valley tech-environment does not just develop technological innovations; it also pursues a political agenda nourished by utopian components, while receiving substantial public subsidies (Alloa, 2019).In a similar, albeit fundamentally different, way, the Chinese state plays a key role in China's technological development (Băzăvan, 2019).

The implementation of automated systems raises the question of what kind of democratic control they are submitted to (De Mul, 2009). Granted, AI-based systems may very well do better than humans by targeting maximum utilitarian optimization, however transferring part of our decision-making process to automated systems means that we consider the sole objective of each decision to be the application of a supposedly rational utilitarian morality. So the problem of moral delegation increases when we have to define the extent to which individuals can be allowed to ignore the decisions of these machine-learning-based systems. We advocate accountability mechanisms and the necessary political control by citizens. An explanation and a certain transparency must accompany the decision that allows individuals to understand why they received such an outcome (Wang et al., 2022). Behind every decision there must be an individual or an organization appointed to be responsible for that decision, and to whom individuals can turn. Those are the minimal conditions for legitimacy.

## 6 Conclusion

Will our values hold? The dangerous property in the "technological state of emergency" in which we live is the impossible return to normalcy (Tesquet, 2021). Invasive technological tools are overturning our conception of morality, but their seemingly inevitable trivialization is keeping us from thinking about it. Technology, through habituation, always ends up going without saying (Heidegger et al., 1986). Technology becomes part of routine. In a 1977 interview with a journalist, Gunther Anders explained the birth of his critique of technology with the bombing of Hiroshima and Nagasaki. He responded to the accusation that technocritical philosophers were merely "panic-mongers". Rather than questioning this accusation, he embraced it (Anders & Greffrath, 2001). We subscribe to this dynamic. One of the most necessary moral tasks today is to make as many people as possible understand that there is cause for concern, or at least for questioning. The delegation of morality to automatic systems must legitimately give rise to debate. As Anders says, "most people are not in a position to create the fear that is needed today. We must therefore help them" (Anders & Greffrath, 2001, pg. 92).

Our ignorance of the moral consequences of the intrusion of moral machines into the spectrum of our actions allows the designers and the users of these automated machines to profit from this system. However, the hype surrounding artificial intelligence needs to be seen for what it is. Machine learning systems are not yet capable of making the final decisions in real-life situations that require global and subjective reasoning capacity. Decision-making aided by such systems based on historical data and learning can be applied in certain fields. However, they lack the empathy that a human being can bring to the ultimate choice. We still need to have individuals in the decision-making process. It is not about delegating responsibility. It is precisely because we do not question the authority we willingly confer on these systems that we suffer the negative consequences.

Here, like in many other areas, we need to turn these technological concerns into political issues. Moral automation, a goal unconsciously pursued by many through the development of automated processes, will be achieved through control. It is therefore crucial to limit the extent of this control in society.

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
