# OpenReview forum: "Are machines automating morality?"
_ICLR.cc/2024/Conference — ICLR 2024 Conference Withdrawn Submission_

### Official Review · Reviewer_CCc6 · 2023-10-26

**Soundness:** 1 poor
**Presentation:** 1 poor
**Contribution:** 2 fair
**Rating:** 3
**Confidence:** 4

**Summary:**

The core concept of the article is the fact that machines automate moral decisions. The authors « try to highlight the prevailing utilitarian ethics found in the tech-centric Silicon Valley culture and its influence on the development of artificial intelligence ». The authors defend the idea that machine learning, nowadays, is driven by utilitarian behavior. The authors examine the ethical implications of machine learning shaping the moral background of society.

**Strengths:**

The topic of abandoning the burden of making moral choices and leaving those choices in the hands of machine-learning models is really important and should be more discussed in the machine-learning community.

**Weaknesses:**

1 – The biggest flaw, in my opinion, is how the article is written on a fine-grained level.

1.1 – The quality of the writing in general needs to be upgraded. For example, some sentences aren’t grammatically valid: « The problem that stems today from machine learning [...] », or « Such was even named the study conducted by researchers […] ».

1.2 - Some references are simply wrong. For example: « According to the Kantian mantra “Live your life as though your every act were to become a universal law” (Kant, 2007) in Critique of Pure Reason, [...] » is wrong, for Kant’s mantra first appeared in his Groundwork of the Metaphysics of Morals. Some sentences are conflicting. For example: « As they do so, a question emerges: Do machines have morals? If so, can they learn about morality? » VS « The problem that stems today from machine learning is that these ”machines” can learn their own morals in an unsupervised way » or « We present the current morality of AI, which [...] » or « These philosophical and psychological questions about the supposed morality of such machines », etc. Another sentence that is false: «In the banking sector, algorithms have largely replaced traditional methods of determining loan eligibility »; algorithms have always been around in insurance; it is the emergence of deep learning or any other black-box in loan eligibility that is new. Finally: « By definition, morality seems to be automatable. Following Kant’s maxim, we could argue that this is the only way to guarantee its value. An action is moral if and only if it is automatable. » The two first sentences argue for the implication « if X is moral, then X is automatable », but the last sentence talks about an if and only if, which, in logic (and philosophy), means both an implication and an inverse implication. But it is not because an action is automatable that it is moral.

1.3 - « By definition, [Kant’s] morality seems to be automatable. » But isn’t all sort of morality automatable? Consequentialist morality could use a proxy for computing the good and bad produced by an event.

1.4 – No rigorous definition of the terms at hand is given. For example, a definition of morality VS ethics was necessary. Or machine learning vs AI vs algorithm, which was only briefly discussed on page 4, after many references to these concepts. Also, there seems to be a lack of understanding of the concepts at hand: « In machine learning, artificial intelligence learns [...] » or «In the banking sector, algorithms have largely replaced traditional methods of determining loan eligibility ».

1.5 – Some sentences lack clarity. For example: « in the banking sector, algorithms have largely replaced traditional methods of determining loan eligibility, leading to moral dilemmas akin to the ”trolley problem” »; what is the link between those two situations? Another example would be « Silicon Valley’s new mantra is WYSIWYG: ”what you see is what you get”. And yet, by brandishing WYSIWYG as its sole political horizon, Silicon Valley delegitimizes in advance any critical reflection on technological innovations and the meaning attached to them (Alloa & Soufron, 2019). » There is a lack of explanation and discussion.

2.1 – Some sentences are vague, questionable, or lack groundings. For example, « Individuals will be more and more dependent on automated systems making moral decisions for them. Therefore, this artificial morality will become the only moral values represented within society. » is purely a slippery slope. Whereas, sentences such as « in the banking sector, [black boxes] have largely replaced traditional methods of determining loan eligibility » aren’t supported. Some sentences such as « We have proved so far that moral values are becoming automated in certain specific sectors » do not leave any place to interpretation, while the claim has not actually been « proven », but simply defended. In social sciences, it is dangerous to state that something has been « proven ».

2.2 – The core of the article is centered around AI being utilitarian. A single citation is used to assess this point, along with a single argument, which is far from being convincing. Plus, the reasoning is doubtful: « consequentialist morality […] has dominated debates and infused current technological developments (Salvat, 2020). As a result, utilitarianism has become the dominant morality in Silicon Valley »: it might have infused technologies, but why assume that because of that it is **dominant** in Silicon Valley? Plus, Silicon Valley is responsible for many technologies that we manipulate in our daily lives, but the example of the loan approval is a good example of something that has huge consequences on the lives of many people, yet does not stem from Silicon Valley. Finally, this last example illustrates how consequentialist vision is not that dominant in AI: we want the models in insurance to be fair and not racist, this criterion being $\textit{a priori}$.

3 - It is unacceptable, in such work, not to discuss « interpretability » or « explainability » in AI, for such topics exactly aim at understanding the decision process of the predictor and currently generate tons and tons of research. Statements such as « [w]ith each moral question addressed by machine learning, we delegate our moral authority. » are not true, for it only (and questionably) holds when it comes to black boxes (some other term that would definitely need to be discussed in such work) in machine learning. Machine learning yielding simple predictors, understandable by human beings could potentially be approved by them.

4 – It is hard to understand the original contribution of the work with regards to the current literature, for most of the ideas in this work are supported by works from the literature; I don’t feel like new or novel thoughts have been brought, or thoroughly defended and argued, therefore questioning the overall contribution of the work.

**Questions:**

See the « Weaknesses » section of the review.

---

> ### Author Response · Authors · 2023-11-22
>
> Dear reviewer, thank very much you for your very detailed comments about our paper.
> Please find our responses to your comments in the following order:
>
> 1 -
>
> 1.1 - You are right, we have tried to make the writing more coherent and to improve the accuracy of grammar and expression.
>
> 1.2 - Kant have developed his categorical imperative in various texts, including Critique of Pure Reason. However, you are right, he developed it for the first time in the Groundwork of the Metaphysics of Morals. We have corrected it.
>
> We have also made corrections about the conflicting sentences. We have introduced the notion of “black box” in the Introduction. However, we have removed the sentence about the banking system. It was confusing and not really relevant to our subject here.
> Finally, we have indeed corrected this implication, the reverse of which does not hold true.
>
> 1.3 - Absolutely. However, we have added the notion of moral "absolutism", which differentiates Kantian morality from consequentialist morality. Kantian morality is valid for everyone at all times, whereas consequentialist morality is concerned with context and purpose.
>
> 1.4 - You are right, there was a lack of definitions, which made the argumentation unclear. We briefly introduced some distinctions in the introduction. However, while these three terms cover different realities, they share the objective of automation (to a greater or lesser extent). It is this automation in general that we are dealing with here, even if we realize that there are differences at various points. We prefer to speak of "automated systems" or "models".
>
> 1.5 - As mentioned above, we have decided to remove the sentence on the banking sector.
> We have extended and integrated into a "Discussion" section the questions surrounding the ideology of Silicon Valley.
>
> 2.1 - "Proving" something is obviously debatable in philosophy. We were using this term in the "philosophical" sense of having tried to prove our point with logical arguments. We're aware that this term can be confusing, so we've modified it and added more of the conditional tense.
>
> 2.2 - Thank you for your detailed and constructive comment. With regard to the utilitarian aspect of Silicon Valley, we are of course aware that many innovations do not originate in Silicon Valley. Our purpose here is to focus on this particular and crucial place when talking about technological innovations.
> We have also briefly compared the Chinese technological ecosystem, in which the ideological aspect plays an important role.
> Finally, from our point of view, insurance models are not designed not to be racist, but to provide the best outcome. They are utilitarian in their functioning. We try to discuss this view in our paperr
>
> 3 - This discussion was indeed missing from our paper. We have therefore introduced this issue first in the introduction and then at various points. We are not experts in the various types of models, and in this paper we are mainly concerned with models that cannot be interpreted or explained. We are aware, though, that the question of interpretability does not apply to simpler models.
>
> 4 - We have reorganized our paper to dedicate part of our work to a discussion in which we seek to further defend a point of view.
>
> We would like to thank you again for all your comments. We hope to have responded satisfactorily.

---

### Official Review · Reviewer_ju5e · 2023-10-31

**Soundness:** 3 good
**Presentation:** 4 excellent
**Contribution:** 3 good
**Rating:** 5
**Confidence:** 2

**Summary:**

This paper provides a philosophical analysis of machine morality, starting from the discussion of if moral values can be gained without explicit human intervention. Then the meaning of moral machines, and the real-world problems that a moral machine might encounter (e.g., the trolley problem). The paper concludes with, "We still need to have individuals in the decision-making process."

**Strengths:**

- This paper is well-written, or very well-written compared to other machine learning conference submissions, with a clear thesis statement and detailed philosophical analysis and references to discuss the topic of moral machines. This topic is also rapidly developing in the current emergent large language model usage in more and more decision-making domains.

- The idea of moral machines, artificial morality and/or automated morality is also a good question posed in this paper. Some previous arguments and experimental evidence (e.g., the constitutional ai or other alignment from ai-feedback work) can also support this.

- This paper gives some interesting points for researchers in this field to focus on, for example, the baby or elderly debate. This kind of question can also be generalized to what morality/values current AI models hold and whether current AI alignment biases too much to WEIRD (Western, Educated, Industrialized, Rich, and Democratic) populations [A].

ref:

[A]. Atari, M., Xue, M. J., Park, P. S., Blasi, D. E., & Henrich, J. (2023, September 22). Which Humans?. https://doi.org/10.31234/osf.io/5b26t

**Weaknesses:**

The most uncertainty I have encountered reviewing this paper is whether this paper fits the ICLR community. Although machine morality is undeniably a significant and rising field in the current learning community, ICLR focuses mostly on technical contributions. I am not sure if this paper actually satisfies ICLR standards (it looks more to me like a Nature Opinion/The New Yorker/or other philosophy conferences), so it might need the Area Chair to judge this paper's suitability. Meanwhile, the paper limit is nine pages (now it's less than six pages), so some additional discussion can be added.

**Questions:**

See strength & weaknesses. I have no further questions besides the fitness to the ICLR community.

---

> ### Author Response · Authors · 2023-11-22
>
> Dear reviewer, thank you very much for your feedback on our paper and your positive appreciation of it. As you pointed out, there was still room for discussion. We have reorganized and lengthened the paper to make this discussion clear.
>
> Our motivation for proposing this paper is the necessary interaction between research in philosophy and research in machine learning on such crucial topics.

---

### Official Review · Reviewer_X5KS · 2023-11-02

**Soundness:** 3 good
**Presentation:** 3 good
**Contribution:** 1 poor
**Rating:** 3
**Confidence:** 3

**Summary:**

The paper reviews the literature around moral decision making and automated systems and says we should think about it more.

**Strengths:**

The authors present a good narrative that is supported by the literature.

**Weaknesses:**

This is just a discussion of the current state of the literature. The review is short and asks many questions without attempting to answer them or present new research directions. As the paper is not at the word limit this shortness is by implication due to the authors believing it was sufficient.

As an example the last paragraph of the main body ends with "perhaps implying a limitation in the learning of moral values." If the authors expanded on this and took a position then there might be a paper. Instead they just summarize existing papers. Section 3 for example is just as summary of the moral machines problem and research.

**Questions:**

What is the research question of this paper? And how is it addressed in a novel way?

---

> ### Author Response · Authors · 2023-11-22
>
> Dear reviewer, thank you for your feedback and comments on our paper. We understand your remark about the probably too centered aspect around a literature review. We have reorganized the paper to give greater prominence to the "Discussion" section.
>
> The thesis of this paper is based on the current and future proliferation of automated systems of all kinds that make moral choices on behalf of humans, whose decisions they replace - partially or totally. We then seek to show that, while these are indeed moral decisions, we must be aware that this moral automation is a choice that responds to human ideological motivations (of control, financial gain...).
> We hope our modified paper is clearer and more personal.

---

> > ### Comment · Reviewer_X5KS · 2023-11-22
> >
> > Thank you for the response and edits to the paper. I am still not clear what your exact research question is and what are the novel contributions. So I will not be changing my score.

---

### Meta-Review · Area_Chair_4L5P · 2023-12-03

**Metareview:**

This is a rather unusual ICLR paper. It reviews the literature around moral decision making and automated systems and says we should think about it more. While all reviwers agree that "moral machines" is an important topic, they also present salient arguments about its suitability for ICLR in its current form such as viewing AI as just being utilitarian. While this might (or not) be the case in Silicon Valley, AI is not just Silicon Valley! Moreover, there is also e.g. machines ethics approache published at FACCT, AIES, Nature MI, NeurIPS, etc. that are unfortunately not discussed. The concerns raised in the reviews should be clarified before publication. We hope that the reviews are useful to you and hope you will find an appropriate forum for this work.

**Justification For Why Not Higher Score:**

Missing related work on machine ethics such as the work on Delphi and Moral Direction.

**Justification For Why Not Lower Score:**

N/A

---

### Decision · Program_Chairs · 2024-01-16

Reject